# Accurate Molecular Diagnosis of Gaucher Disease Using Clinical Exome Sequencing as a First-Tier Test

**DOI:** 10.3390/ijms22115538

**Published:** 2021-05-24

**Authors:** Stefania Zampieri, Silvia Cattarossi, Eleonora Pavan, Antonio Barbato, Agata Fiumara, Paolo Peruzzo, Maurizio Scarpa, Giovanni Ciana, Andrea Dardis

**Affiliations:** 1Regional Coordinator Centre for Rare Diseases, University Hospital of Udine, 33100 Udine, Italy; stefania.zampieri@asufc.sanita.fvg.it (S.Z.); silvia.cattarossi@asufc.sanita.fvg.it (S.C.); pavan.eleonora@gmail.com (E.P.); paolo.peruzzo@asufc.sanita.fvg.it (P.P.); maurizio.scarpa@asufc.sanita.fvg.it (M.S.); giovanni.ciana@asufc.sanita.fvg.it (G.C.); 2Department of Clinical Medicine and Surgery, Federico II University Hospital, 80131 Naples, Italy; abarbato@unina.it; 3Pediatric Unit, Regional Referral Center for Inherited Metabolic Disease, University of Catania, 95123 Catania, Italy; fiumara@policlinico.unict.it

**Keywords:** *GBA*, clinical exome sequencing, MLPA

## Abstract

Gaucher disease (GD) is an autosomal recessive lysosomal disorder due to beta-glucosidase gene (*GBA*) mutations. The molecular diagnosis of GD is complicated by the presence of recombinant alleles originating from a highly homologous pseudogene. Clinical exome sequencing (CES) is a rapid genetic approach for identifying disease-causing mutations. However, copy number variation and recombination events are poorly detected, and further investigations are required to avoid mis-genotyping. The aim of this work was to set-up an integrated strategy for GD patients genotyping using CES as a first-line test. Eight patients diagnosed with GD were analyzed by CES. Five patients were fully genotyped, while three were revealed to be homozygous for mutations that were not confirmed in the parents. Therefore, MLPA (multiplex ligation-dependent probe amplification) and specific long-range PCR were performed, and two recombinant alleles, one of them novel, and one large deletion were identified. Furthermore, an MLPA assay performed in one family resulted in the identification of an additional novel mutation (p.M124V) in a relative, in trans with the known p.N409S mutation. In conclusion, even though CES has become extensively used in clinical practice, our study emphasizes the importance of a comprehensive molecular strategy to provide proper *GBA* genotyping and genetic counseling.

## 1. Introduction

The *GBA* gene (OMIM #606463) encodes for the lysosomal hydrolase acid b-glucocerebrosidase (GCase), a 497-amino acid membrane glycoprotein of 65 kDa responsible for the breakdown of the glycolipid glucosylceramide to ceramide and glucose, in association with the cofactor Saposin C. Bi-allelic pathogenic variants in the *GBA* gene cause Gaucher disease (GD), a rare inherited disorder in which the deficient activity of GCase leads to the progressive storage of glucosylceramide and other glycosphingolipids within lysosomes, resulting in multiorgan system disease [1]. In addition, the presence of heterozygous *GBA* gene variants represents a major risk factor to develop Parkinson’s disease (PD), as observed in a large cohort of PD patients, particularly in those with early-onset [2,3,4,5,6,7]

The *GBA* gene consists of 11 exons and 10 introns, covering approximately 7.6 kb on chromosome 1q21. This region is composed of seven genes and two pseudogenes within an 85-kb genomic portion, probably derived from duplication events through evolution [8,9].

Approximately 16 kb downstream of the functional *GBA* gene, a nonfunctional highly homologous pseudogene (*GBAP1*) that shows a similar exon–intron organization [10] has been identified. Both nonreciprocal and reciprocal recombination events occur due to the high degree of homology and the close proximity between *GBAP1* and *GBA*, promoting the formation of complex recombinant alleles [11,12].

To date, more than 500 different disease-causing *GBA* mutations have been identified in patients affected by GD, including single base changing, splicing alterations, deletions, insertions, and recombinant rearrangements (Human Gene Mutation Database, http://www.hgmd.org 21 May 2021). Specifically, more than 20 recombinant alleles have been characterized; among them, RecNcil and Recdelta55 represent the most common alleles either alone or in combination with additional point mutations [13,14,15]. Large deletions have also been identified [16,17,18]. Most of the recombinant pathogenic mutations usually occur between intron 8 and the 3′untranslated region where the gene and pseudogene share 98% of sequence homology.

Clinical exome sequencing (CES) is a powerful genetic test consisting of a massively parallel sequencing strategy, enabling the rapid identification of variants in disease-causing genes. Nowadays, with the decrease in the cost of sequencing, CES has become affordable and is extensively used for the molecular diagnosis of mendelian disorders [19].

However, CES and next-generation sequencing approaches, in general, present technical limitations in detecting mutations such as large rearrangements, structural variants and copy number variations (CNVs), epigenetic modifications, mutations in repetitive or GC-rich regions, or mutations in highly homologous genomic regions [20].

Indeed, the existence of the *GBAP1* pseudogene and the presence of recombination events may cause misalignments in the generated reads, making the genetic analysis of GD patients using NGS techniques challenging. A highly reliable and efficient pipeline to deal with *GBA* sequencing, in which the misalignment of reads was avoided by including a specific long-template *GBA* amplification for library preparation and a specific procedure for data analysis, has been developed [21]. However, the molecular analysis of *GBA* using CES has not been examined.

In the current study, we set out a strategy to achieve the complete genotyping of eight GD patients, integrating CES with other molecular techniques, such as MLPA (multiplex ligation-dependent probe amplification) and long-range specific PCR. MLPA is a semi-quantitative molecular strategy used for the detection of copy number variants in specific genes using in a single multiplex PCR-based reaction. Long-range PCR is needed to unambiguously amplify the *GBA* gene, avoiding pseudogene interference. This approach, previously applied as guidelines for analyzing other genes with corresponding pseudogenes or highly homologous genes [22,23], led us to the identification of two recombinant alleles, one of them new, and a whole gene deletion. In addition, further investigations allowed us to diagnose the father of one patient in our cohort, as affected by GD, and to identify and characterize a novel missense mutation, p.M124V (M85V).

In conclusion, this study emphasizes the limitations of CES and the importance of a comprehensive molecular approach to obtain accurate *GBA* genotyping for proper prognosis and reliable genetic counseling to GD families, as well as to support the inclusion of the *GBA* gene in NGS-based assays.

## 2. Results

### 2.1. Clinical Exome Sequencing (CES) and Variant Identification

CES was performed on eight unrelated patients affected by GD. The average clinical exome coverage 10% quantile was 90 ± 7.1X, while the *GBA* gene coverage was 100% at ≥20X. CES analysis led us to fully genotype five patients (GD#4-GD#8) who were compound heterozygotes for already reported point mutations in the *GBA* gene (Table 1). Co-segregation analysis performed by Sanger sequencing confirmed the presence of mutations in the relatives.

Three patients were revealed to be homozygotes for the following mutations: p.L483P (L444P) for GD#1 and p.N409S (N370S) for GD#2 and GD#3. The carrier status of the GD#1 and GD#3 parents was explored by Sanger sequencing and, surprisingly, only one parent was revealed to be a carrier in each family. Segregation analysis was not possible for GD#2, because the parents were unavailable; thus, further studies were needed for this patient to confirm the presence of a true homozygous status.

The results obtained from segregation analysis in GD#1 and GD#3 relatives strongly suggested that these patients would carry the identified point mutation in compound heterozygosity with a recombinant or a deleted allele, nonidentified by CES. To confirm this hypothesis, an MLPA assay was performed, revealing the presence of large structural changes of *GBA* in all three GD patients (Figure 1A–C). However, this technique does not differentiate between deletions or recombination. Therefore, to further characterized these alleles, the whole *GBA* gene was analyzed by Sanger sequencing. A long-range PCR to selectively amplify the active gene was performed [21,24] (Figure 1D–F).

MLPA on GD#1 patient was characterized by a profile consistent with the presence of a CNV covering the whole gene from exon 1 to exon 10, in addition to the missense mutation p.L483P, which generated a lower signal of exon 10 as it is located within the binding region of the MLPA probe (Figure 1A). Probes matching the *GBA* 5′ untranslated region and flanking gene CKS1B were normal (data not shown). The long-template *GBA*-specific PCR amplification of exons 1–5 and 5–11 followed by sequencing revealed the presence of single bands of normal molecular weight and the p.L483P (L444P) mutation in the homozygous state (Figure 1D). Thus, we hypothesized the presence of a large deletion in the *GBA* gene covering the whole gene. The MLPA analysis of the parent confirmed the maternal origin of the deleted allele (Figure 2A,B). Despite several attempts to characterize the deletion junction, the deletion extent is currently unknown.

In patient GD#2, MLPA showed a CNV alteration from at least exon 3 to exon 10, while exon 1 was normal (Figure 1B). *GBA*-specific PCR amplification of the genomic portion spanning from exon 1 to exon 5 revealed a lower abnormal amplicon, in addition to the wild-type amplicon (Figure 1E). Sequencing of the low-molecular-weight amplicon showed the presence of a novel recombinant allele extending from exon 2 up to the 3′ untranslated region (AH006907.2: g.2094con10194).

It is important to highlight that we were unable to sequence the parents of GD#2; therefore, without a complete molecular analysis, this patient would have been misdiagnosed.

MLPA performed in patient GD#3 revealed an alteration in the CNV at least from exon 3 to exon 9 (Figure 1C). Again, the presence of abnormally sized PCR products after specific *GBA* amplification was observed. Sequencing analysis showed the presence of a recombinant allele extending from exon 2 to nucleotide c.1448 in exon 10 (Figure 1F) (AH006907.2: g.1942_7319con10042_13806) [25].

Co-segregation analysis of the recombinant allele was extended to GD#3 parents via MLPA assay, showing that it had a maternal origin.

Unexpectedly, the GD#3’s father, who is a carrier of the p.N409S (N370) mutation, also showed an abnormal pattern in the MLPA assay corresponding to the probe matching the exon 4 region. Sanger sequencing revealed the presence of a novel variant of unknown significance, c.370A > G, (p.M124V (M85V)), in trans with the p.N409S (N370S) mutation (Figure 2C,D). Therefore, considering the hypothesis that the father could also be affected by GD, we decided to better define the impact of p.M124V (M85V) on the GCase activity, to provide more complete and accurate information to the family before starting a complete diagnostic workout in the father.

### 2.2. Functional Analysis of p.M124V (M85V) Variant

The functional analysis of missense variant p.M124V on GCase activity and protein expression was evaluated by the in vitro expression of the mutant pCDNA3-*GBA* M124V construct in Hek293 cells in a *GBA* null background (Hek293-*GBA* KO) [26]. Hek293-*GBA* KO cells were also transfected with a pCDNA2-*GBA* WT plasmid and pCDNA3-ALD unrelated construct as a positive and negative control, respectively.

After 48 h, Hek293 cells were harvested and analyzed for *GBA* protein expression and enzymatic activity. No changes in the viability of transfected cells were observed.

As shown in Figure 3A, cells expressing the mutant construct retained 20% of the GCase activity compared to cells expressing the WT construct, despite a comparable GCase protein abundance (Figure 3B).

Based on this result, GCase activity was assessed in the patient’s leucocytes. A GCase residual activity of 0.7 nmol/mg/h (normal range: 4.5–18.3) confirmed the diagnosis of GD.

The patient was 56 years old at the time of the diagnosis. He had hypertension, hypercholesterolemia, a moderate (KDOQI stage 3) kidney chronic disease and an IgG kappa monoclonal gammopathy of undetermined significance (MGUS). He did not show any other clinical sign or symptoms such as hematological disorders, hepatosplenomegaly, or bone pain suggestive of Gaucher disease. In addition, the biochemical markers evaluated, such as chitotriosidase, resulted in the normal range. However, it is worth noting that this patient was a heterozygous carrier of a duplication of 24 bp in exon 10 of the chitotriosidase gene (CHIT1), which abolished plasma chitotriosidase activity [27].

## 3. Discussion

The presence of the *GBAP1* pseudogene makes the *GBA* molecular analysis and GD diagnosis challenging. Indeed, long-range PCR using specific primers represented the gold standard strategy to characterize the *GBA* gene, and it is widely used in clinical practice for the molecular diagnosis of GD [24,28,29,30,31]. Hence, the detection of CNVs and *GBA*-*GBAP1* recombinant alleles turned out to be challenging and time-consuming in the GD diagnostic routine, and it represents a major pitfall in GD mutational analysis. Considering the broad spectrum of clinical phenotypes, the molecular genotyping of GD patients is extremely important not only for diagnosis confirmation, prognostic, and therapeutic purposes, but also for the proper counseling of patients and family members.

The CES approach has been increasingly used to identify disease-causing variants in monogenic and complex disorders [19,32]. However, although software has been developed for CNV detection generated from whole-exome or whole-genome sequencing data, CNV identification is limited. Moreover, as previously reported, NGS data pipelines for the identification of *GBA* mutations are unable to detect recombinant variants, unless using a modified workflow [21].

In the present study, eight patients affected by GD were successfully genotyped using several complementary approaches such as CES, MLPA, and Sanger sequencing of long-range specific PCR products. Five GD patients, compound heterozygotes for pathogenic point mutations, were correctly genotyped by CES. In fact, Sanger sequencing and co-segregation analysis confirmed that potentially pathogenic variants were inherited from heterozygous parents.

However, three patients carrying either recombinant alleles or large deletions were revealed to be apparent homozygotes for point mutations. Distinguishing true homozygosity from apparent homozygosity is essential for proper genotyping, especially when the parents are not available, as in the case of patient GD#2. In fact, the impact of an apparently homozygous genotype may be quite critical with respect to the predicted phenotype and the response to mutation-specific therapies such as molecular chaperones.

These results confirm the poor performance of CES in detecting the CNV, and they highlight the need to use complementary strategies, such as MLPA, to identify this type of mutation. Among the *GBA* CNV, we identified a large deletion covering the whole gene in association with the deleterious mutation p.L483P (L444P) in GD#1. A complete deletion of the *GBA* gene has been previously reported [16]. Gross deletions are more difficult to identify than partial deletions, mostly because they will not produce abnormal bands on PCR amplification and could not be detected even after long-template specific PCR. Therefore, to prove the presence of such deletions, MLPA studies should be performed, especially considering that family studies are not always possible.

In addition, we identified two complex alleles in GD#2 and GD#3 patients that were the result of *GBA*-*GBPA1* rearrangements. Both contain functional gene sequences up to exon 2 and pseudogene sequences downstream. These alleles would eventually lead to the synthesis of a nonfunctional GCase protein. The recombinant allele g.2094con10194 has not been previously reported.

Analyzing the co-segregation pattern of recombinant alleles in GD#3’s family, we identified a novel missense variant, p.M124V (M85V), in trans with the p.N409S (N370S) mutation in the father. In vitro expression studies showed that the p.M124V (M85V) mutant retains 20% of enzymatic activity when compared to wild-type *GBA*. Therefore, GD#3’s father was revealed to be a compound heterozygous for p. M124V (M85V) and p.N409S (N370S) mutations. Although he was asymptomatic, the enzymatic activity performed on peripheral blood leukocytes confirmed the diagnosis of GD in this relative.

The absence of clinical signs of GD in this patient is consistent with the mild pathogenetic nature of both the p.N409S (N370S) and the novel p.M124V (M85V) mutation.

The data presented here confirmed major limitations of CES in the detection of either the *GBA* recombinant allele or the deletion in GD patients and the need of using additional and complementary tests to prevent mis-genotyping. In fact, CNVs are rarely tested in routine screenings and might remain unidentified in some patients.

In particular, our results indicate that the use of methods suitable for the detection of CNV due to deleted or recombinant alleles became mandatory when mutations in homozygosity were identified by CES. Moreover, this approach should be considered when using NGS for the study of conditions associated with mutations in different genes, including *GBA*, such as Parkinson’s disease [33], dementia with Lewy bodies [34], lysosomal diseases [35], and inherited platelets disorders [36]. Finally, other genes that are characterized by the presence of high homologous sequences or pseudogenes might benefit from the adoption of this integrated molecular strategy.

## 4. Materials and Methods

### 4.1. Samples

Genomic DNA was collected from eight patients affected by GD and family members, whenever possible. Written informed consent was obtained from adult subjects or from both patients or parents/guardians on behalf of the minors involved in this study. None of the patients have consanguineous parents. The clinical diagnosis of GD was confirmed via biochemical assay, showing decreased levels of GCase activity in peripheral blood leukocytes or culture skin fibroblasts.

### 4.2. Molecular Analysis

Genomic DNA was extracted from whole peripheral blood samples using the Qiamp Blood extraction mini kit (Qiagen GmbH, Hilden, Germany).

CES analysis was performed using the Clinical Exome solution by the SOPHiA™ Genetics kit (Sophia Genetics SA, Saint-Sulpice, Switzerland), on a Miseq platform (Illumina), following the manufacturer’s instruction. Raw data processing, and variant calling and annotation were conducted using SOPHiA™ DDM v5.7.5 (Sophia Genetics SA, Saint-Sulpice, Switzerland) bioinformatics pipelines. The CES panel includes 4493 clinically relevant genes. The identified mutations were confirmed in the parents whenever possible and were validated by specific long-range PCR amplification, using primers designed to discriminate between *GBA* and *GBAP1* sequences (GenBank AH006907.2) [28], followed by automated Sanger sequencing (ABI Prism 3500xl genetic analyzer, Applied Biosystems, Foster City, CA, USA).

### 4.3. Multiplex Ligation-Dependent Probe Amplification (MLPA) Analysis of GBA Gene

The screening for *GBA* copy number changes due to CNV or complex rearrangements was performed using SALSA MLPA probemix P338-B1 (MRC-Holland, Amsterdam, The Netherlands) containing 9 *GBA* probes targeting exons 1, 3, 4, and 6–10 and the 5′UTR region of transcript variant 1 (NM_000157.3). Briefly, 100 ng of each genomic DNA was denatured at 98 °C for 10 min. Then, the MLPA probemix was added before an overnight hybridization at 60 °C. Probes were ligated at 54 °C for 15 min and then PCR-amplified. PCR products were separated on an ABI PRISM^®^ 3500XL Genetic Analyzer (Applied Biosystems, Foster City, CA, USA) and data were analyzed using Coffalyser.Net software (MRC-Holland).

### 4.4. Mutation Nomenclature

The mutation nomenclature follows the current HGVS guidelines (Human Genome Variant Society, http://www.hgvs.org/mutnomen 21 May 2021), numbering the A of the first ATG translation initiation codon as nucleotide +1 (RefSeq NM_000157.3) [37,38]. Traditional amino acid residue numbering, which excludes the first 39 residues corresponding to the signal peptide, has also been provided within parentheses and without the p. prefix.

### 4.5. Site-Directed Mutagenesis

PCR-based site-directed mutagenesis (Stratagene, Cedar Creek, TX, USA) was used to introduce the novel mutation p.M124V in the complete wild-type *GBA* cDNA cloned in the pcDNA3 vector, following the manufacturer’s instructions. The oligonucleotides containing the specific mismatches were Primer Forward: GGATTTGGAGGGGCCGTGACAGATGCTGCTG and Primer Reverse: CAGCAGCATCTGTCACGGCCCCTCCAAATCC. The mutant construct was completely sequenced to confirm that no mutations other than p.M124V were introduced by the mutagenesis procedure.

### 4.6. Cell Culture and Transient Transfection

Hek293 *GBA* knocked out cells [26] were grown in Dulbecco’s modified Eagle’s medium (DMEM) supplemented with 10% fetal calf serum, 2 mM of glutamine and 50 mg/mL of penicillin/streptomycin (Gibco, Paisley, UK) at 37 °C in a humidified atmosphere enriched with 5% (*v*/*v*) CO_2_. For transfection, cells were seeded into 6-well plates in medium without antibiotics and were transfected with either 2 µg of the wild-type or mutant *GBA* construct using Lipofectamine 2000 (Invitrogen Carlsbad, CA, USA) according to the manufacturer’s instructions. As a negative control, cells were transfected with pCDNA3-ALD (Adrenoleukodystrophy protein), a peroxisomal transmembrane protein not correlated with GD disease, to guarantee similar transfection conditions among samples. After 48 h, the cells were harvested, and the cellular extracts were assessed for GCase activity and Western blot analysis.

### 4.7. Enzyme Activity Assay

The GCase enzymatic activity was measured using the fluorogenic substrate 4-methylumbelliferyl- β-d-glucopyranoside (4MU-β-d-glu) (Sigma-Aldrich, St. Louis, MO, USA). The protein concentration of the samples was determined by the Lowry method. Briefly, 10 μL containing 10 μg of protein was incubated for 3 h with 10 μL of the substrate, 5 mM in acetate buffer, 0.1 M, pH 4.2, at 37 °C. The reaction was terminated by the addition of carbonate buffer into 0.5 M, pH 10.7, of stop solution, and the fluorescent product was quantified using a fluorimeter (SPECTRAmax Gemini XPS, Molecular Devices, San Jose, CA, USA) at an excitation wavelength of 365 nm and emission of 495 nm. The standard curve was built using 4-methylumbeliferone (4-MU) as a standard, and for each sample, the nanomoles of product generated by a milligram of protein in one hour was calculated. The residual enzymatic activity was expressed as the percentage of wild-type values.

### 4.8. Western Blotting Analysis

For Western blot analysis, cell pellets were lysed using TNN Buffer (Tris-HCl 100 mM pH 8, NaCl 250 mM, NP40 0.5%) supplemented with protease inhibitors (Sigma-Aldrich, St. Louis, MO, USA). After samples sonication and centrifugation, the protein concentration was calculated using the Bradford protein assay (BioRad, Hercules, CA, USA), following manufacturer’s protocols. Thirty micrograms of total protein extracts from transfected Hek293 cells were resolved on a 10% gradient mini-protean TGX pre-cast gel (BioRad, Hercules, CA, USA) and were transferred onto nitrocellulose membranes (Biorad, Hercules, CA, USA). After blocking within 5% blotting-grade blocker (BioRad, Hercules, CA, USA) in PBS-T (0.1% Tween 20 in PBS), the blotted membranes were incubated overnight at 4 °C with the following primary antibodies: *GBA* 2E2 (WH0002629M1, Sigma-Aldrich, St. Louis, MO, USA) and Vinculin (E1E9V, Cell Signaling Technology Inc., Danvers, MA, USA). Membranes were then washed, incubated with the appropriate secondary antibody (Dako Agilent, Santa Clara, CA, USA) for 1 h at RT, and developed with SuperSignal West Dura/Pico reagents (Thermo Fisher Scientific, Waltham, MA, USA). The signals were normalized to those obtained for actin. Blots were quantified by using a Uvitec Cambridge Imaging system (UVITEC, Cambridge, UK).

## 5. Conclusions

In summary, our study supported the mandatory use of an integrated approach of CES, MLPA and long-range PCR for the molecular analysis of the *GBA* gene. Our investigation led us to fully genotype eight GD patients, and to the identification of novel recombinant alleles, a large deletion, and a novel missense mutation, p.M124V (M85V).

The strategy described here might provide important insights for future molecular analysis of patients affected by GD.

## Figures and Tables

**Figure 1 ijms-22-05538-f001:**
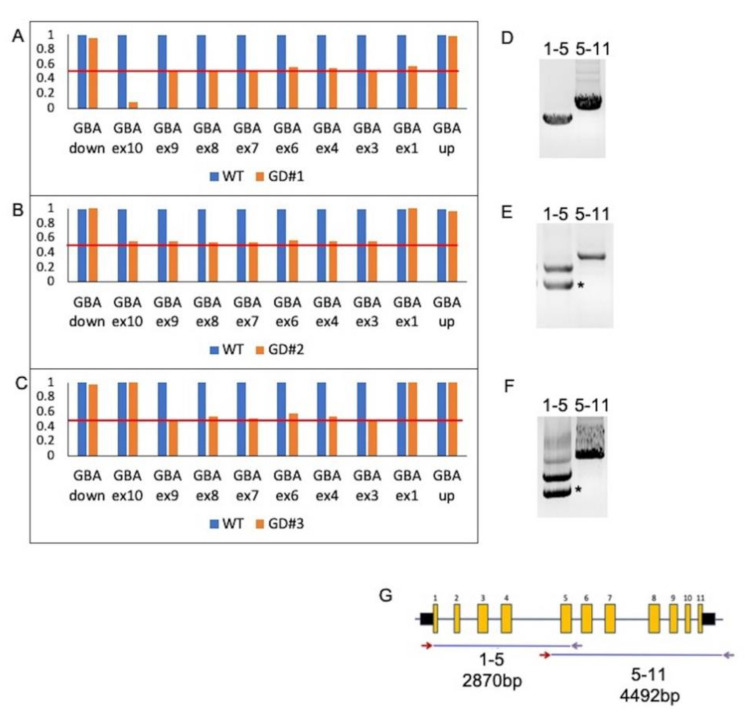
MLPA analysis conducted on GD#1, GD#2, and GD#3 patients. (**A**–**C**) Normalized bar charts of *GBA* gene copy number patterns in GD patients (orange bars) and healthy control (blue bars). Data were normalized to those of healthy controls. (**D**–**F**) PCR amplicons of exon 1 to 5 and of exon 5 to 3′ untranslated region of *GBA* gene of patients GD#1 (**D**), GD#2 (**E**), and GD#3 (**F**). (**G**) Schematic representation of *GBA* long-range PCR amplicons. Exonic and intronic structures of the *GBA* gene are indicated by yellow boxes and grey lines, respectively. A forward primer (red arrow) specific to the *GBA* gene and an unspecific reverse primer (purple arrow) were used to selectively amplify the *GBA* gene and to avoid pseudogene contamination. * PCR amplification of recombinant allele.

**Figure 2 ijms-22-05538-f002:**
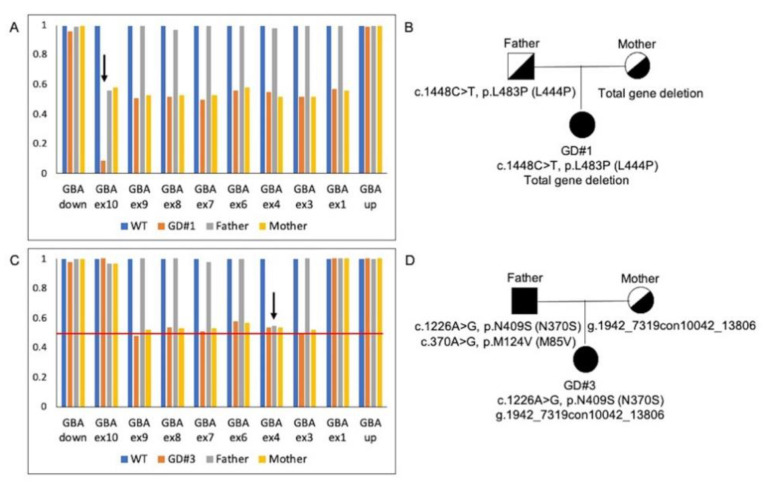
Co-segregation analysis performed by MLPA in GD#1 and GD#3’s family. (**A**,**C**) Normalized bar charts of *GBA* gene copy number patterns in the proband (orange bars), the mother (yellow bars) and the father (grey bars). Data were normalized to those of healthy controls (blue bars). (**A**) GD#1’s family co-segregation analysis. The black arrow indicates the absence of the peak corresponding to the *GBA* exon 10 probe in the proband. The father presented a half-reduced exon 10 relative peak height, while the mother displayed a half-reduced relative height of all *GBA* exons. (**B**) Family pedigree showing genotypes in GD#1. (**C**) GD#3’s family co-segregation analysis of the AH006907.2: g.2094con10194 recombinant allele. The black arrow indicates a half-reduced relative peak height corresponding to the *GBA* exon 4 probe in GD#3’s father. The mother presented a half-reduced relative height of exons 3 to 9. (**D**) Family pedigree showing genotypes in GD#3.

**Figure 3 ijms-22-05538-f003:**
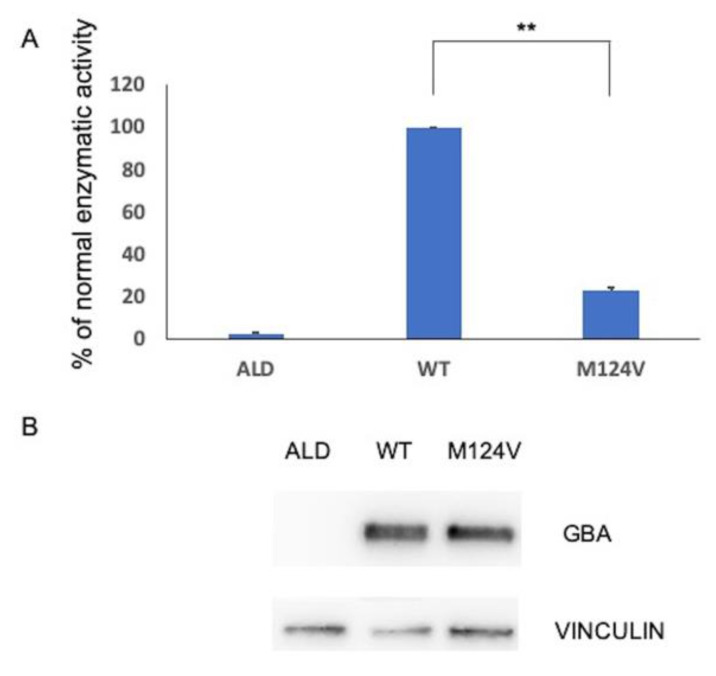
Functional analysis of the novel p.M124V (M85V) missense mutation. (**A**) GCase activity in Hek293-*GBA* KO cells transiently transfected with the pCDNA3-*GBA* WT (WT) plasmid and pCDNA3 *GBA* M124V (M124V). Values are expressed as the percentage of wild-type activity and represent the mean ± SEM of three independent experiments. ** *p* < 0.001. pCDNA3 ALD was used as a negative control. (**B**) Western blot analysis of WT and mutant *GBA* protein expressed in Hek293-*GBA* KO cells.

**Table 1 ijms-22-05538-t001:** *GBA* gene mutations identified in GD patients.

Patient	Allele 1	Allele 2
GD#1	c.1448T>C, p.L483P (L444P)	Total gene deletion
GD#2	c.1226A>G, p.N409S (N370S)	AH006907.2:g.2094con10194
GD#3	c.1226A>G, p.N409S (N370S)	AH006907.2:g.1942_7319con100042_13806
GD#4	c.1049A>G, p.H350R (H311R)	c.1226A>G, p.N409S (N370S)
GD#5	c.413delC, p.P138Lfs*62 (P99Lfs*62)	c.1226A>G, p.N409S (N370S)
GD#6	c.1226A>G, p.N409S (N370S)	c.1448T>C, p.L483P (L444P)
GD#7	c.475C>T, p.159W (R120W)	c.1226A>G, p.N409S (N370S)
GD#8	c.508C>T, p.R170C (R131C)	c.1448T>C, p.L483P (L444P)

*GBA* cDNA accession number NM_000157 (NP_000148). A novel mutation is indicated in red. The parentheses include traditional codon numbering beginning 39 codons downstream from the first ATG.

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
