# Peer review of "Accurate Molecular Diagnosis of Gaucher Disease Using Clinical Exome Sequencing as a First-Tier Test"

_ijms, 2021, doi:10.3390/ijms22115538_

Round 1
Reviewer 1 Report
I think the study is appropriate for publication in this journal. The paper supports the mandatory use of an integrated approach of CES, MLPA and long-range PCR for the molecular analysis of GBA gene. They fully genotype eight GD patients and the identification of a novel recombinant alleles, a large deletion and a novel missense mutation, p.M124V (M85V). Overall, this strategy described here might provide important insights for future molecular analysis of patients affected by GD. However, some points of the paper need to be better described:
figure 3: why you used mean and SD instead of mean and SEM if experiments are independent? Are 3 independent experiments or 3 replicate? In case are only replicate, you have to do 3 independent experiments and show their SEM to obtain statistically acceptable results.
lane 178: what VN mean?
lane 19: explicit MLPA the first time you use it
lane 303 : “As a negative control, cells were transfected with pCDNA3-303 ALD (Adrenoleukodystrophy protein), a peroxisomal transmembrane protein not correlated with GD disease.” why you use this type of negative control and not the native plasmid? you have to explain it or put references
lane 315 = do you used a calibration line to calculate the specific activity as nmol/mg/h? you have to write it in the methods
lane 318 : write how did you process samples before the run on the gel
Reviewer 2 Report
The present manuscript is a nice documentation of the use of the Multiplex Ligation-dependent Probe Amplification (MLPA) analysis, following clinical exome sequencing (CES), to disclose the genotypes of three Gaucher disease patients containing recombinant alleles or deletions. Using the MLPA technique, the authors were also able to identify that the father of one of the patients was a GD patient. It seems that the combination of CES with MLPA is ideal in GD for identification of complex alleles.
The manuscript should be published following minor corrections as specified below.
COMMENTS
A short description of the MLPA technique is recommended in the introduction. The authors should add the complete name of MLPA the first time the abbreviation is being used in the abstract and in the introduction.
The authors refer to the GD associated gene as GBA. There is a GBA2 gene and a GBA3 gene, so why GBA and not GBA1 (specially, when there is GBA1P)? Also, genes should appear in italics.
Line 51: “RecNcil and Recdelta being the most common either alone or in combination with additional point mutations [13,14]”. The reference for recNciI should be: Eyal et al, Gene, 96 (1990) 277. For recdelta55, please add: Filocamo et el, Human Mutation 20 (2002) 234.
Line 66: “making e the genetic analysis of GD patients using NGS …”. Omit the e.
Line 97: “Instead, three patients resulted homozygotes for the following mutations: p.L483P. Omit : instead.
Lines 102-103: The connection between the paragraph ending in line 102 and the paragraph starting in line 103 is not clear. It seems that something is missing between them.
Line 103: Please change heterozygosis to heterozygosity.
Lines 112-117: The authors mention that: “Probes matching on GBA 5’ untranslated region and flanking gene CKS1B were normal (data not shown). However, PCR amplification and sequencing reveal the presence of a unique PCR product and the presence of the p.L483P (L444P) mutation in homozygous state (Figure 1D)…..”
Please change “on” to “the”. It is not clear which is the unique PCR product in Figure 1D. The authors also wrote that: “MLPA analysis of parent’s confirmed the maternal origin of the deleted allele“. It will be nice to see the MLPA result of the parents’ DNA.
Lines 131-133: …“showed the presence of a recombinant allele extending from exon 2 and nucleotide c.1448 in exon 10”. Should be: from exon 2 to nucleotide c. 1448 in exon 10.
Line 143-144: “Co-segregation analysis of the recombinant allele was extended GD#3 parents by MLPA assay”.. Should be:..was extended to GD#3 parents…
Line 146: …”showed an abnormal pattern in the MLPA assay involving to exon 4 region. Edit language.
Line 148: ..The authors mention that ”… defining the possible pathogenic impact of p.M124V (M85V) variant was crucial to determine whether the father was affected by GD”.
Determining whether the father was a GD patient was performed by measuring activity of lysosomal glucocerebrosidase.
Lines 168-: “This result suggests that although pathogenetic, the p.M124V variant might be considered as a mild mutation.”
Since the authors did not use in their transfection assay any known mild or severe mutations, it is difficult conclude about the severity of the M124V mutation.
Line 213: Please correct “chaperons” to “chaperones”.
Line 242: please change “homozygosis” to “homozygosity”.
